# Polyurea Thickened Lubricating Grease—The Effect of Degree of Polymerization on Rheological and Tribological Properties

**DOI:** 10.3390/polym14040795

**Published:** 2022-02-18

**Authors:** Max Jopen, Patrick Degen, Stephan Henzler, Bastian Grabe, Wolf Hiller, Ralf Weberskirch

**Affiliations:** 1Faculty of Chemistry and Chemical Biology, Technical University of Dortmund, Otto-Hahn-Str. 6, 44227 Dortmund, Germany; max.jopen@tu-dortmund.de (M.J.); bastian.grabe@tu-dortmund.de (B.G.); wolf.hiller@tu-dortmund.de (W.H.); 2Carl Bechem GmbH, Weststraße 120, 58089 Hagen, Germany; degen@bechem.de (P.D.); stephan.henzler@bechem.de (S.H.)

**Keywords:** shear rheology, organogel, grease, polyurea, rheological properties, tribological properties, molecular mass, SRV-tribometer, flow limit, lubrication, anti-wear (AW), extrem-pressure (EP)

## Abstract

Lubricating greases based on urea thickeners are frequently used in high-performance applications since their invention in 1954. One property that has so far been neglected in the further development of these systems due to their low solubility and the resulting difficulty of analysis, is to better understand how the degree of polymerization affect such polyurea lubricating systems. In this work, we prepared three different oligo- or polyurea systemswith different degrees of polymerization (DP) and investigated the influence of DP on rheological and tribological properties. The results showed that the DP has an influence on the flow limit in rheology as well as on the extreme pressure (EP) and anti-wear (AW) properties as examined by tribology measurements. By optimizing the DP for a thickener system, comparable EP and AW properties can be achieved through the use of additives. The DP showed an increasing influence on the flow limit. This could reduce damage to rolling bearings due to lateral loading at rest. Therefore, modifying the DP of the polyurea systems shows similar effects as the addition of external additives. Overall, this would reduce the use of additives in industrial applications.

## 1. Introduction

Lubricating greases are an important component for machines and from today’s point of view it is impossible to imagine life without them. Their primary purpose is to reduce or completely prevent friction between machine parts. They are also employed to provide protection against corrosion, as well as against external influences such as humidity [1]. Lubricating greases consist of three components: a base oil, a thickener and additives. In commercially available greases, the proportion of base oil is 65–95%, the thickener proportion 3–30% and the proportion of additives 0–10% [2]. The majority of greases are metal soap greases, however, also graphite [3], polyethylene [4], PTFE [5], and ureas [6,7] are used as metal-free thickeners with urea based thickeners being the most important group. Chemically speaking, lubricating greases often belong to the organogels [8].

Urea greases were first invented by Swaken et al. [9] in 1954 and are characterized by their excellent physical properties, including high anti-oxidative stability, high temperature resistance and high mechanical stability [10,11]. Due to their physical properties, they are often used in rolling bearings for high-performance applications. Since the discovery of urea-based thickeners until today, the focus has been on the development and improvement of diurea greases [10,12,13,14]. In 1965, Traise et al. [15] firstly reported the production of an oligourea based grease. In 1970, Dreheret al. [16] investigated the influence of aryl and alkyl groups on the physical properties of tetraurea based greases. In 1986, toluene-2,4-diisocyanate (TDI) based urea greases for the first time [17] and in 1993 the influence of processing parameters such as temperature during the production of polyurea on the performance of such greases were reported [18]. In 1995, Root et al. [19] presented the first fibrous polyurea grease, which showed an improved performance as well as a high water resistance. The fibrous structure is similar to the fibrous structures that as also reported for soap greases.

Since 2010 there has been a continuous development and improvement of urea-based greases. This involved improving the processing in the production of urea greases [20,21], investigations of the influence of the base oil [22,23,24,25,26], variation of the reaction components [27] and development of additives to further improve the lubricant properties [28,29,30,31,32,33,34,35,36,37,38].

Interestingly, only little attention was paid to the effect of the thickener structure although the thickener makes up to 30 wt% of the final lubricants. Maksimova et al. [27] investigated toluene-2,4-diisocyanate (TDI) as a reaction component and its effect on the physical properties of urea greases. Liu et al. [39] reported for the first time that the number of urea units could correlate with the properties of the grease. However, a major challenge was a correlation of molar mass of the urea thickener with rheological and tribological properties of the final lubricant. Another topic of increasing interest is the sustainability of the lubricants. Typically, the starting materials for urea chemistry, diisocanates and diamines, are based on petrochemical base chemicals. In recent years, however, great efforts have been made to develop new routes to bio-based aliphatic diamines, such as HDA and ODA, by metabolic engineering [40]. In addition, Covestro has succeeded in making aniline available from renewable raw materials for the first time with the help of a fermentation process [41,42]. This means that in the future both 4,4′-methylenediphenyl diisocyanate (MDI), the most important aromatic diisocyanate, and the precursor methylenedi(phenyldiamine) MDA could be available from bio-based raw materials making urea-based thickeners also attractive components for more sustainable lubricants.Here, we report the synthesis of lubricants with three types of polyurea based thickeners composed of 4,4′-methylenediphenyl diisocyanate (MDI) (1) as the diisocyanate component and methylene di(phenyl diamine) (MDA) (2a), 1,8-octyl diamine (ODA) (2b), 1,6-hexyldiamine (HDA) (2c) as the diamine comonomer with a theoretical degree of polymerization ranging from 5 to 13 (Figure 1). Molar mass determination of the polyurea thickeners were accomplished after soxhlet extraction, protonation with a strong acid and subsequent ^1^H NMR end group analysis. This allowed an unambiguous correlation of polyurea degree of polymerization and rheological and tribological properties of the final lubricants. 

## 2. Materials and Methods

For all greases, PAO6 base oil (INEOS Manufacturing GmbH, Cologne, Germany) was used, as well as methylene diphenyl diisocyanate (MDI, 98%) Alfa Aesar (Kandel, Germany), 4,4′-diaminodiphenylmethane (MDA, >98%) from TCI (Zwijndrecht, Belgium), 1,6-hexamethylene diamine (98%) from Sigma Aldrich (Munich, Germany), 1,8-octamethylene diamine (ODA, 99%) from Sigma Aldrich (Munich, Germany), and *n*-stearyl amine (>85%) from TCI (Zwijndrecht, Belgium).

### 2.1. Grease Synthesis

All polyurea greases were produced directly by an in-situ polyaddition reaction in the PAO6 base oil. An ULTRA TURRAX^®^ Tube Drive with ST-20 mixing vessel was used to ensure effective homogenization during polymerization. The monomers were heated until they melted and the oil was heated to the same temperature (approximately 95 °C). Then the oil was added to the reaction vessel and the molten diisocyanate was added to the oil while stirring. Afterwards the amine mixture (diamine and monoamine) was added under stirring (5000 rpm) and stirring was continued for another 2 min. Typically, the reaction mixture showed a gel-like consistency after only a few seconds. Conversion was controlled by FTIR spectroscopy and the disappearance of the isocyante C-N band at 2270 cm^−1^. If conversion was incomplete further monoamine was added to the reaction mixture until no isocyanate band was present. After the polymerization was finished, the thickener proportion was optimized for each grease. Therefore, the grease was centrifuged at 5000 rpm for two minutes to remove excess oil as supernatant. The greases were then homogenised by means of a roller (EXAKT 50) with 10 µm gap and a rotation of 600 rpm. Air bubbles were removed in a vacuum drying oven at 40 °C and reduced pressure.

For the production of all greases 20 mL base oil was used. To calculate the theoretical amount of polymeric thickener, 8.6 g was chosen. The required amounts of diisocyanate, diamine and monoamine are therefore calculated via the stoichiometric ratio depending on the theoretical repeat units according to the Carothers equation [43]. As theoretical repeat units for all systems 13.5, 11, 9, 7.5, 6 and 5 were chosen. This results in a stoichiometric ratio of diisocyanate:diamine:monoamine of 1:0.84:0:16, 1:0.80:0.20, 1:0.75:0.25, 1:0.69:0.31, 1:0.60:0.40 and 1:0.50:0.50 respectively.
(1)$$\mathrm{DP}=\frac{1+r}{1+r-2xr}\;\mathrm{with}\;r=\frac{N_{A0}}{N_{B0}+2N_{B^{\prime }0}}$$

Equation (1): Carothers equation with *r* as ratio of the monomeres, *x* as conversion and DP as the degree of polymerization [43].

### 2.2. ^1^H NMR Analysis of the Polyurea Thickener

The molar mass of the thickeners was determined by ^1^H NMR spectroscopy on a 400 MHz Avance HD-III Nanobay (Agilent Technologies, Ratingen, Germany). The manufactured greases were subjected to a soxhlet extraction with *n*-pentane to isolate the polymeric thickener. For ^1^H NMR analysis the dried thickener powder was dissolved in concentrated sulfuric acid (98%). A closed capillary with deuterium oxide (D_2_O) was used as reference for the chemical shift of the signals during the measurement. The signal of the sulfuric acid (11–12 ppm) was suppressed during the measurement by means of solvent suppression. To determine the molar mass, the integral of the methylene end group signal (0.65 ppm) was referenced to 6 protons and the methylene bridge signal (3.83 ppm) was divided by the number of protons per repeat unit. ^1^H NMR signals for the MDI-MDA system: (400 MHz, D_2_O) δ = 7.5–7.1 ppm (m, 16 H), 3.9–3.8 (m, 4 H), 3.1 (s, 4 H), 1.3 (s, 4 H), 1.1 (s, 60 H), 0.7 (t, 6 H). The number of protons shall be multiplied by the number of repeat units.

^1^H NMR signals for the MDI-ODA system: (400 MHz, D_2_O) *δ* = 7.7–7.1 ppm (m, 16 H), 3.9–3.8 (m, 2 H), 3.2 (m, 8 H), 1.4 (m, 8 H), 1.1 (m, 68 H), 0.6 (t, 6 H). ^1^H NMR signals for the MDI-HDA system: (400 MHz, D_2_O) *δ* = 7.7–7.1 ppm (m, 16 H), 3.9–3.8 (m, 2 H), 3.2 (m, 8 H), 1.4 (m, 8 H), 1.1 (m, 64 H), 0.6 (t, 6 H).

### 2.3. Penetrometer and NLGI Analysis

The NLGI class was determined by unworked penetration according to DIN ISO 2137, using a PNR 10 Penetrometer (Petrotest, Dahlewitz, Germany). The penetration of a cone into the grease at room temperature was measured and assigned to the respective NLGI class according to the immersion depth.

### 2.4. Rheology

Oscillation measurements were performed on an MCR 302 rheometer (Anton Paar, Ostfildern, Germany) to determine the flow limit according to DIN 51810-2 at 25 °C. A plate-plate geometry with a diameter of 25 mm was used with a shear deformation of 0.01% to 100% (41 data points), constant angular frequency of 10 rad/s and constant gap size of 1 mm. The flow limit is determined about the intersection point between storage (*G′*) and loss modulus (*G″*).

### 2.5. Tribology

Load level tests were performed on an SRV III tribometer (Optimol Instruments, Munich, Germany). Oscillation measurements (linear oscillation) according to ASTM D5706 were performed. The load was gradually increased over 2 h (50 N constant for 5 min) until 1200 N. The geometry used was a ball-plate system with 17.4 mm diameter made of 100Cr6 steel (lapped plate). 

## 3. Results

### 3.1. Synthesis of the Lubricants and Determination of the Molar Mass

Three types of greases were produced by in-situ polymerization of methylene diphenyl diisocyanate (MDI) with three diamino compounds, i.e., 4,4′-diaminodiphenylmethane (MDA), 1,6-hexamethylene diamine (HDA) or 1,8-octamethylene diamine (ODA) in PAO6 as base oil. *n*-Stearyl amine was added to control the final molar mass of the oligomers according to the Carothers equation. Theoretical values for the degree of polymerization were between 5 and 13.5 for all three polyurea types (see also Table A1, Table A2 and Table A3). The progress of the polyaddition reactions was monitored by FT-IR to determine any residual isocyanate groups that were quenched with additional *n*-stearyl amine. To correlate molar mass of the polyurea thickeners with the rheological and tribological properties of the final lubricant we extracted the polymer from the base oil by soxhlet extraction. The resulting polymers were white solids that displayed little solubility in organic solvents due to crystallinity and strong intermolecular hydrogen bonds [44]. A possibility to increase solubility of semicrystalline polymers like polyamides is based on the break-up of these hydrogen bonds with strong acids [45]. By applying this approach to our polyureas, endgroup analysis by ^1^H NMR spectroscopy could be carried out in concentrated sulfonic acid (Figure 1). To ensure a comparable determination of the molar mass, the integral of the signal of the methylene end group (0.65 ppm) was referenced to six protons and the signal of the methylene bridge (3.83 ppm) was divided by the number of protons per repeat unit. 

The molar masses determined from the ^1^H NMR’s are summarized for each urea series in Table A1, Table A2 and Table A3. Typically, the values determined by ^1^H NMR endgroup analysis were slightly lower than the theoretically expected ones for all three polyurea series. The deviation in the degree of polymerization was more pronounced for the longer oligomers or polymers with 10 or 13.5 repeating units whereas the shorter oligomers or polymers with 5 to 9 repeating units were in good agreement with the theoretical values. The reason for the limitation to build up longer oligomers or polymers is probably due to the highly viscous lubricant mixture that is formed in-situ during the polyaddition reaction and the intermolecular interactions and thus reduced accessibility of the oligomer or polymer endgroups. When considering the degrees of polymerization, the number of urea units per repeat unit must be taken into account. Each repeat unit contains 2 urea units. A degree of polymerization of 5 thus already means that the polymer chains contain 10 urea units on average.

### 3.2. NLGI Class Determination

The NLGI class is a measure of the strength or stiffness of a grease. For application purposes, lubricating greases of the same NLGI class are mostly compared with each other, since the strength can be regarded as a parameter of the stability or the crystalline portion of the thickener. The NLGI classes of the greases were determined with static penetration. As can be seen from Table A4 two observations were made. All greases showed a larger penetration with increasing degree of polymerization suggesting that the greases are getting softer when the molar mass of the polymeric thickener increases. The greases of the MDI-MDA series displayed NLGI classes between 2–4 whereas the other two lubricating grease systems MDI-HDA and MDI-ODA showed NLGI classes between 0–1 and are significantly softer than the MDI-MDA greases. Thus, the effect of the chemical structure of the polymeric thickener has a much stronger effect on the static penetration behavior and thus on the final NLGI class than the degree of polymerization within an oligourea or polyurea series.

### 3.3. Rheology

In a previous publication [46], we have already shown oscillation rheology measurements of bio-based polyurea greases and the influence of the chemical thickener structure on the flow limit. The oscillation measurements were carried out to determine the yield point and the flow limit of the different greases at 25 °C. The yield point here indicates the end of the linear viscoelastic range (LVE) [47]. Oscillation measurements in this range reversibly deform the gel. However, since lubricating greases are usually not only reversibly deformed in applications, the focus of the measurements was the determination of the flow limit which indicates the shear stress *τ*_F_ or deformation *γ*_F_ limit above which a viscoelastic solid no longer behaves in a predominantly elastic manner. The flow limit can be referred to as the elastic limit or as transition point from viscoleastic solids to viscoelastic liquids and is determined by the intersection point between storage (*G′*) and loss modulus (*G*″). As mentioned before, it is possible to determine the intersection of *G′* and *G″* for a percentage deformation *γ*_F_ or the measured shear stress *τ*_F_ at this deformation (Figure A1). The shear stress *τ*_F_ corresponds to the absolute flow limit, while the deformation *γ*_F_ is equivalent to a relative flow limit. The aim was to correlate the previously determined molar masses and the resulting chain length of the polyurea with the flow limit. However, the correlation is based on two problems: On the one hand all polyurea greases were optimized with regard to their thickener content. The final lubricants displayed between 5 and 20 wt% thickener content depending on the series investigated and the degree of polymerization within each series (see Table A5, Table A6 and Table A7), however, no clear trends were observable.

In order to exclude the influence of the thickener content, five tetraurea (DP = 2) greases with MDI-MDA thickener were produced with a thickener content range of 5 to 18 wt% and examined with regard to their absolut (*τ*_F_) and relative (*γ*_F_) flow limit (Figure 2, Table A8). Based on these results, it can be clearly shown that the absolute flow limit (*τ*_F_) is dependent on the thickener content, while the relative flow limit (*γ*_F_) shows independent behavior. For further investigation, the flow limit was therefore determined on the basis of deformation in order to create comparable relative values. Since this fact is far from obvious, an explanation could be found by plotting measured shear stress *τ* against belonging deformation *γ* (Figure 3). Here, too, a comparable curve progression with increasing thickener content can be seen. Only the shear stress range on the Y-axis shifts. From 18% by weight, there is a change in the shape of the curve. 

It is possible to determine the flow limit from this plot (Figure A2) [47] which is independent of the thickener concentration. Mathematically, this can be approximated by applying the Windhab model [48]. However, since the Windhab model represents a strong simplification for a grease with oligomeric or polymeric thickener, it is a highly simplified approximation.

*τ*_y_ − *τ*_F_ mark yield point (*τ*_y_) and flow limit (*τ*_F_) of the sample. According to Metzger [48] an evaluation of the flow limit by means of this application is possible, but scientifically it is too inaccurate. Therefore, a determination of the flow limit via the intersection of Log *G′* and Log *G″* is recommended. The Windhab model describes the relationship between flow limit *τ*_F_, shear stress *τ*, deformation *γ* and intrinsic viscosity *η* for a suspension, and thus the transition from the quiescent to the yield state (Figure A1).
(2)$$\tau =\tau_{y}+\left(\tau_{F}-\tau_{y}\right)\cdot \left[1-\exp\left(-\frac{\gamma }{\gamma^{*}}\right)\right]+\eta_{\infty }\cdot \gamma$$

Equation (2): Mathematical relationship of rheological quantities using the Windhab model [48].

($\eta_{\infty }$ = intrinsic viscosity at high deformation. *γ** = deformation at flow limit)
(3)$$\gamma^{*}=\tau_{y}+\left(\tau_{F}-\tau_{y}\right)\left(1-\left(\frac{1}{e}\right)\right)$$

Equation (3): Mathematical relationship of rheological quantities at deformation at the flow limit using the Windhab model [48].

For polymers, the intrinsic viscosity is defined by Equation (4).
(4)$$\eta_{\infty }=K\cdot M^{\alpha }$$

Equation (4): Intrinsic viscosity for polymers [49].

*M*^α^ depends on the solubility of the polymers. Since this decreases with increasing thickener concentration and the grease becomes firmer with otherwise constant parameters, the intrinsic viscosity depends directly on the thickener concentration. For lubricating greases or organogels, this is a major simplification. A mathematical explanation is provided by the Windhab model via Equations (1) and (2). Equation (1) shows that the absolute flow limit in Pascal depends on the intrinsic viscosity. However, the deformation *γ** at the flow limit is independent of the intrinsic viscosity.

Further, the determined relative flow limit *γ*_F_ of the 18 grease systems under investigation were plotted against the previously determined degree of polymerization (DP) (Figure 4). The degree of polymerization was chosen for the correlation, since it represents a measure of the mobility of the polymer chains [46] and thus allows a better comparison of the structurally different systems. All three system series reveal a linear increase of *γ*_F_ with increasing DP, however, they differ in their slope. Whereas the MDI-HDA and MDI-ODA series indicate a similar slope of increasing *γ*_F_ with increasing DP, the slope for the MDI-MDA series is significantly flatter. Although it cannot explain the difference based on the chemical structure of the polyurea thickener an increasing *γ*_F_ may lead to delayed performance for the MDI-HDA and MDI-ODA based greases at higher DP. For LCP’s, it is known that more aromatics in the polymer backbone can lead to lower chain mobility [50]. Therefore, both the MDI-HDA and the MDI-ODA systems probably have a higher but similar chain mobility. The higher chain mobility allows for a better match of chain alignment to increasing shear stress. With increasing DP, this mobility increases further. This results in a higher slope for these two systems and an increasing slope for each series.

In the penetration measurements described previously, it was shown that the greases become softer with increasing DP, thus possibly having a lower NLGI class. This could be seen as a loss of stability of the gel structure. The penetration was performed as a static measurement on the grease at rest, plastic deformation should occur at this point. In order to investigate such a loss of stability, the constant elastic modulus *G′* and the loss modulus *G*″ in Pascal for the LVE were plotted against the DP for all three polyurea systems in the following (Figure 5). In addition, the determined penetration depth, from which the NLGI class results, was correlated with the moduli.

Physically, a higher *G′* in the LVE range can be assumed to result in a more stable network [51]. Figure 6 shows that for the MDI-ODA and the MDI-HDA system *G′* decreases with increasing DP. For these systems, the penetration measurement results surprisingly correlate with a decrease in stability of the network. This behavior is not seen for the MDI-MDA system. In some cases, the system even shows significantly higher *G′* values at higher DP and thus a more stable network. This system must therefore be based in part on different mechanisms of physical crosslinking or conformation of the superstructure than for the other two systems. It is usually assumed that the crosslinking points of an urea gels are formed by crystalline regions [52]. In this context, a larger DP should lead to a smaller crystalline fraction, thus decreasing the stability. For MDI-MDA, however, this phenomenon does not occur. This could be an indication that amorphous structures can also act as crosslinking points of urea gels.

### 3.4. Tribology

In addition to rheology, the tribological behavior of a lubricant is of great importance for application development. In their application, lubricating greases are frequently subjected to high forces and pressures, as well as changing loads. Two important properties that are therefore necessary for the evaluation of lubricating greases are the extreme pressure behavior (EP) and the anti-wear behavior (AW) [53]. One way to investigate both EP and AW properties in one is to perform a load level test on a vibrating friction wear tribometer (SRV) [54]. Such a test was therefore carried out for all 18 systems in the following. The evaluation of the measurement results of this test was carried out according to four different evaluation criteria: The load level at which lubrication failure occurs, the run time of the measurement until lubrication failure occurs, the total run time until system failure, and the area under the friction coefficient function.

The load level at which lubrication failure occurs was determined by the occurrence of a frictional impact, a so-called stroke. This evaluation criterion enables the EP properties of the system to be considered. The higher the load level reached, the higher the EP properties of the system. The load levels achieved for each system using this evaluation approach are shown in Figure 6. According to this criterion, for all three thickener series, an influence of the DP on the achieved load level and, consequently, on the EP property of the system can be seen. For the MDI-MDA system, this influence only occurs for shorter polymer chains. For longer chains, the maximum step load of the test of 1200 N is always reached. For this system, higher DP thus lead to better EP properties with regard to this evaluation criterion. The MDI-ODA and MDI-HDA show similar behavior as before in the rheological tests. Here, the achieved load level first increases with decreasing DP before reaching its maximum. Subsequently, the achieved load level drops again with further decreasing DP. The shortest polymer chains at a DP of 5 again exhibit a higher load level. This deviating behavior is due to the different chemical structure of the systems. The MDI-MDA system series exhibits lower chain mobility than the other two system series. With increasing DP, the crystallization potential of the system increases and thus the stability. The other two system series exhibit similar and higher chain mobility due to their chemical structure. Depending on the chain length, different folding mechanisms of the polymer chains can thus occur. This may result in an optimum at a certain DP.

An evaluation according to the reached load level, however, does not cover all parameters of such a test. Experimentally, a load level is kept constant for 5 min and then increased by 50 N. The load level reached therefore does not provide any information on whether the system gave up when changing to a new load level, during a load level or at the end of a load level. Therefore, the runtime of the measurement at which a stroke occurs controlled. This criterion is accordingly based on the same data basis and should therefore show a correlation. As this evaluation criterion classifies the examined systems more precisely within their reached load level, it can be regarded as a correction criterion of the reached load level and thus also provides information on the EP properties of a system. The runtime achieved for each system using this evaluation approach are shown in Figure A2.

A look at the data shows that the results correlate directly with the previous evaluation criterion. For the MDI-MDA and MDI-ODA systems, this evaluation does not provide any new information. However, for the MDI-HDA system, no clear optimum could be determined previously with respect to the DP. However, the correction criterion shows that for a DP of 8.0 and 7.3 in this system, both samples reach the same load level, but the shorter polymer chain achieves a longer run time. This correlates with the observations for the comparable MDI-ODA system.

One problem with these two evaluation criteria is that no information about the type of event can be obtained here. If there is only a brief increase in friction, it is not possible to speak of a complete failure of the lubricating effect. The system recovers after the event and the measurement continues. The friction and thus the wear has therefore not yet reached a level within this event at which irreparable damage to the lapped measuring plate occurs. The total runtime of the measurement until the measuring system is switched off is therefore another important evaluation criterion that also provides information on the AW properties of the grease system. The results according to this evaluation criterion are shown in Figure 7.

Taking into account the constant 5 min per load level, the maximum runtime of the load stage test is 130 min. For the MDI-MDA system, it can be seen that only the two systems with a DP of 4.0 and 3.0 do not reach the maximum runtime of the test. Thus, irreparable damage already occurs at 85 and 93 min, respectively. Regarding the AW properties, larger DPs seem to be favored here. A similar picture emerges when comparing the reached load levels. It is noticeable that the load level reached was determined by a stroke at 66 and 72.5 min. However, the actual occurrence of the irreparable damage takes place at a later time. A short-term recovery of the grease system can thus be assumed. An opposite effect can be observed for the MDI-ODA system. Only the samples with a DP of 10 and 9.5 do not reach the maximum runtime here. Thus, shorter DP seem to be favored for AW properties. The system failure lies in the same time frame as the occurrence of the strokes during the determination of the reached load level. A recovery does not occur here. For the MDI-HDA system, all samples reached the maximum run time. This is particularly noticeable as none of the specimens reached the maximum load level of 1200 N without a significant event. This suggests good recovery behavior for this system. This effect appears to be DP independent.

Another evaluation criterion for AW properties is based on the friction coefficient over the entire measurement. However, since this value is influenced by varying degrees of deflection during the stroke, this would result in a variable with varying degrees of error. Therefore, the area under the friction coefficient function was used. In order to keep this the same for samples with stroke, the areas from stroke were normalized to the highest point of the stroke up to the maximum running time. The results of this assessment are shown in Figure 8.

This evaluation shows that all systems except for two samples, produce comparable friction values. The results correlate with the evaluation in terms of the total runtime of the measurement. Thus, the AW properties of these systems appear to depend much less on DP than the EP properties do. 

## 4. Discussion

Both the rheological and tribological investigations showed an influence of the DP on the different measured values or criteria. The chemical significance of these data have already been discussed in the respective chapter. However, two questions that are still not conclusively clarified: How can the rheological data be causally correlated with the tribological data and what do the respective results mean for the application as lubricating grease systems?

For rheology, the flow limit *γ*_F_ was found to increase with increasing DP. The flow limit represents an elasticity limit of the respective system, but what does this mean for the application? In the application, the lubricating greases are mostly exposed to conditions above the flow limit. Elastic deformation is therefore negligible. The flow limit therefore has little or no influence on the direct performance of a system in the application and can therefore not directly correlated with the tribological results. Elastic deformation can, however, occur in one case: In bearings at rest that are subjected to influences such as vibration from the outside due to transport or other machines. In the latter case, a higher elastic limit would be desirable, as this would prevent possible damage due to lateral loading. Larger DPs would therefore be desirable here.

For tribology, a dependence of the EP properties on the DP was found. Depending on the chemical structure, there seems to be an optimum at a certain DP. Up to this optimum, an improvement of the EP properties with increasing DP can be seen. The optimum is probably due to a more favorable conformation of the polymer chains and thus the superstructure of the physical network. For the AW properties, there was hardly any dependence on the DP. For the MDI-HDA system, no influence was found for the DP produced. For the MDI-MDA system, a significant influence is shown for small DP. For the MDI-ODA system, there was little influence for large DP. In terms of application, AW and EP properties play an important role. Often these properties are improved by the use of additives. At this point, it should be mentioned that only pure base greases, without any additives were investigated within the scope of this publication. By knowing the DP influence, these properties can be controlled depending on the system and the use of additives can be reduced.

Even though a direct correlation of the rheological and tribological data is not possible, a common feature is evident for both series of measurements: In both series of measurements, the results are influenced by the chemical structure and DP of the thickeners. Both parameters can be combined into one influence, the chain mobility. For all measurements, the trend is that the influence of the chemical structure appears to be greater for smaller DP. The chain mobility is therefore dominated by the chemical structure. For larger DP, the influence of the chemical structure seems to be smaller and the chain mobility is significantly influenced by the DP.

However, it is not possible to make a fundamental recommendation regarding the DP, since this is both system- and application-dependent. Optimization for the desired properties must therefore be carried out separately for each system and changes in NLGI class, AW and EP properties as well as changes in the flow limit must be taken into account.

## 5. Conclusions

Here we have investigated the effect of molar mass and chemical structure of three different oligourea series on different properties of the resulting greases. Therefore, three different series of oligourea thickener were prepared by polyaddition reaction of methylene diphenyl diisocyanate (MDI) and three diamines, i.e., 4,4′-diaminodiphenylmethane (MDA), 1,6-hexamethylene diamine (HDA) and 1,8-octamethylene diamine (ODA).

The final greases contained between 5 and 20 wt% of the oligourea thickener with a degree of polymerization between 3 and 10.5. Rheological and tribological analysis of the different greases showed that strain deformation and NLGI class determination displayed a strong dependence on the degree of the polymerization of the oligourea thickener as well as the molecular structure of the thickener used. The results allow for the first time a clear correlation of different molar mass of the oligourea thickener with the rheological and tribological properties such as extreme pressure behavior (EP) and the anti-wear behavior (AW) of the resulting greases. Future work will be devoted to an in debt analysis of the structure of the grease and the urea thickener within the grease as a dependence of DP and molecular structure to better understand the tribological behavior described in this manuscript.

## Data Availability

The data presented in this study are available on request from the corresponding author.

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
