# Peer review of "Polyurea Thickened Lubricating Grease—The Effect of Degree of Polymerization on Rheological and Tribological Properties"

_polymers, 2022, doi:10.3390/polym14040795_

Round 1
Reviewer 1 Report
In this manuscript, the authors described the synthesis of lubricants with three types of polyurea based thickeners, and the influence of DP on rheological and tribological properties was investigated. This research is well designed and organized, so it could be published after minor revisions.
- The authors used three diamine comonomers (MDA, ODA and HDA) and one diisocyanate (MDI) for polyurea synthesis, any other diisocyanates were investigated besides MDI?
- The purity of the materials listed in “2. Materials and Methods” should be provided.
- The scale bar of the SEM recordings in Fig. S23-25 should be added.
- Summary or Conclusion is missing in the manuscript.
Reviewer 2 Report
Apply an error bar and a smooth line on all plots.
An advanced discussion supporting conclusions about the role of molecular structure, DP and interactions on chain mobility and macroscopic properties, would be an advantage.
